# Three-Dimensional Radiological Assessment of Ablative Margins in Hepatocellular Carcinoma: Pilot Study of Overlay Fused CT/MRI Imaging with Automatic Registration

**DOI:** 10.3390/cancers13061460

**Published:** 2021-03-23

**Authors:** Yasunori Minami, Tomohiro Minami, Kazuomi Ueshima, Yukinobu Yagyu, Masakatsu Tsurusaki, Takuya Okada, Masatoshi Hori, Masatoshi Kudo, Takamichi Murakami

**Affiliations:** 1Department of Gastroenterology and Hepatology, Faculty of Medicine, Kindai University, 377-2 Ohno-Higashi Osaka-Sayama, Osaka 589-8511, Japan; tekatyuu@yahoo.co.jp (T.M.); kaz-ues@med.kindai.ac.jp (K.U.); m-kudo@med.kindai.ac.jp (M.K.); 2Department of Radiology, Faculty of Medicine, Kindai University, 377-2 Ohno-Higashi Osaka-Sayama, Osaka 589-8511, Japan; y-yagyu@med.kindai.ac.jp (Y.Y.); mtsuru@dk2.so-net.ne.jp (M.T.); 3Department of Radiology, Kobe University Graduate School of Medicine, 7-5-2 Kusunoki-cho, Chuo-ku, Kobe, Hyogo 650-0017, Japan; okabone@gmail.com (T.O.); horimsts@med.kobe-u.ac.jp (M.H.); murataka@med.kobe-u.ac.jp (T.M.)

**Keywords:** ablative margin, hepatocellular carcinoma, image fusion, radiofrequency ablation, treatment assessment

## Abstract

**Simple Summary:**

Recent advances in fusion imaging technology have made it easier to visualize and estimate ablative margins. This study was conducted to assess the clinical feasibility of a computed tomography (CT)/magnetic resonance imaging (MRI) fusion application for evaluation of the ablative margin in radiofrequency ablation (RFA) for hepatocellular carcinoma (HCC). Seventeen patients developed local tumor progressions (LTPs) due to wrong initial evaluations of technical success through a side-by-side comparison, and we reevaluated the ablative margins using the CT/MRI overlay fusion application. Eight patients were categorized into grade C (margin-zero ablation) and nine patients into grade D (existence of residual HCC). LTP occurred in re-graded C patients within 4 to 30.3 months (median, 14.3 months), and in re-graded D patients within 2.4 to 6.7 months (median, 4.2 months) (*p* = 0.006). Overlay fused CT/MRI imaging can allow us to evaluate HCC ablative margin three-dimensionally with high accuracy.

**Abstract:**

Background: We investigate the feasibility of image fusion application for ablative margin assessment in radiofrequency ablation (RFA) for hepatocellular carcinoma (HCC) and possible causes for a wrong initial evaluation of technical success through a side-by-side comparison. Methods: A total of 467 patients with 1100 HCCs who underwent RFA were reviewed retrospectively. Seventeen patients developed local tumor progressions (LTPs) (median size, 1.0 cm) despite initial judgments of successful ablation referring to contrast-enhanced images obtained in the 24 h after ablation. The ablative margins were reevaluated radiologically by overlaying fused images pre- and post-ablation. Results: The initial categorizations of the 17 LTPs had been grade A (absolutely curative) (*n* = 5) and grade B (relatively curative) (*n* = 12); however, the reevaluation altered the response categories to eight grade C (margin-zero ablation) and nine grade D (existence of residual HCC). LTP occurred in eight patients re-graded as C within 4 to 30.3 months (median, 14.3) and in nine patients re-graded as D within 2.4 to 6.7 months (median, 4.2) (*p* = 0.006). Periablational hyperemia enhancements concealed all nine HCCs reevaluated as grade D. Conclusion: Side-by-side comparisons carry a risk of misleading diagnoses for LTP of HCC. Overlay fused imaging technology can be used to evaluate HCC ablative margin with high accuracy.

## 1. Introduction

Imaging-guided ablative therapies have become fundamental in the treatment of hepatocellular carcinoma (HCC), and have proven to be competitive with surgery in terms of overall survival in cases of single nodules less than 2 cm [1,2,3,4]. However, higher local recurrences have been demonstrated in HCC patients treated with percutaneous ablation therapy [5,6,7,8,9], reaching higher than 40% during 2–3 years of follow-up [10,11]. It is well known that the local tumor progression (LTP) rate differs markedly depending on whether or not a 5 mm ablative margin is secured [12]. However, ablation therapy including radiofrequency ablation (RFA) does not always achieve a 5-mm safety margin for HCC in clinical practice [13,14,15]. Moreover, more precise evaluation of treatment response is fundamentally difficult. The therapeutic response has conventionally been evaluated by comparing axial images of computed tomography (CT) or magnetic resonance imaging (MRI) obtained before and after ablation therapy, usually in a side-by-side manner [12]. Ablation zones are usually measured with the eye by the differences in relative distances between intrahepatic landmarks such as blood vessels or cysts. Therefore, this approach is limited by an inability to assess the ablative margin precisely at millimeter level. 

Important issues in the treatment response of ablation therapy are evaluations of the absence of tumor enhancement and the ablative margin [12]. Contrast-enhanced ultrasound (US) can depict signals from microbubbles with a very slow flow and is a useful tool for depicting small residual or locally recurrent HCC [16,17]. However, consistent and accurate assessments of the ablative margin by CEUS is limited because the tumor boundary may not be clearly identified on US after ablation. Meanwhile, fusion imaging allows the physician to directly understand the intrahepatic anatomy between imaging modalities, including CT, MRI and US. Fusion imaging of CT or MRI using a workstation is effective for the assessment of therapies such as RFA for HCC. Although previous studies have demonstrated CT/MRI fusion imaging to be useful in assessing the safety margin for early response of RFA [18,19,20,21], these applications employed a conventional side-by-side comparison, and operation of the application software was sometimes complicated. 

Recent advances in CT/MRI fusion imaging have made it easier to visualize and estimate ablative margins. New applications can overlay pre- and post-RFA images, with the tumor image radiologically projected onto the ablation zone [22,23]. For example, two concentric circles on CT containing the tumor border within the ablative low-density area would be shown in a good response case (Figure 1). In addition, the application can adjust the tumor location automatically between pre- and post-ablation images. The aim of this study was to assess the clinical feasibility of a CT/MRI fusion imaging application for evaluation of the ablative margin in RFA for HCC and possible causes for a wrong initial evaluation of technical success through side-by-side comparison. 

## 2. Results 

### 2.1. Patient Characteristics

Between January 2014 and September 2016, we performed percutaneous RFA under ultrasound guidance for the treatment of 1100 HCC tumors in 467 patients (364 men and 103 women; age range, 35–93 years; mean age, 69.2 years). Despite grade A (absolutely curative) or B (relatively curative) judgments of their early responses referring to contrast-enhanced images obtained in the 24 h after ablation, LTP occurred in 17 patients until August 2017. We enrolled the 17 patients in this study and reevaluated the ablative margins of RFA using the CT/MRI overlay fusion application with automatic registration. Seventeen patients (13 men, 4 women; age range, 58–79 years; mean age ± SD, 68.3 ± 5.7 years) with 17 HCCs were analyzed. Baseline features of the study population are summarized in Table 1. Fifteen patients had liver cirrhosis of Child-Pugh class A and the remaining two had Child-Pugh class B. The maximum diameter of HCC before RFA ranged from 0.7 to 2.7 cm (mean ± SD, 1.4 ± 0.5 cm). The recurrent HCCs were distributed as 4 left lateral, 3 left medial, 5 right medial, 6 right lateral or none in segment 1. One patient had not previously been treated for HCC. Twelve patients had previously been treated by RFA at other sites in the liver. The remaining four patients had previously been treated by RFA (*n* = 2) or transcatheter arterial chemoembolization (TACE) (*n* = 2) at the same site. The mean long and short axis diameters of each ablation zone were 3.2 cm (SD, 0.9; range, 1.6–5.4 cm) and 2.5 cm (SD, 0.5; range, 1.4–3.3 cm). The differences in the long axis diameter between the tumor and ablation zone ranged from 0.9 to 2.8 cm (mean, 1.7 cm). 

### 2.2. Three-Dimensional Assessment of Ablative Margins Using Overlay Fused CT/MRI Imaging 

Although the initial classification of the 17 LTPs had been grade A (absolutely curative) (*n* = 5) and grade B (relatively curative) (*n* = 12) with conventional side-by-side visual comparison of pre- and post-RFA images (Figure 2), the reevaluation with radiological overlay fusion altered the response categories to eight grade C (margin-zero ablation) and nine grade D (existence of residual HCC). Most discrepancies in the assessment of the RFA response could be mainly caused by reactive hyperemia covering tumor enhancement (*n* = 9), a misleading impression induced by a large non-enhancing area (*n* = 7) and the limit in the sagittal margin assessment (*n* = 1). Notably, the nine residual HCCs re-graded as D were all covered and obscured by periablational hyperemia enhancement (Figure 3). The size of LTPs ranged from 0.6 to 1.6 cm (median, 1.0 cm). For the whole cohort, LTP occurred within 2.4 to 30.3 months (median, 6.6 months). LTP occurred in the eight patients re-graded as C within 4 to 30.3 months (median, 14.3 months), and in the nine patients re-graded as D within 2.4 to 6.7 months (median, 4.2 months). There was a significant difference between these two groups (*p* = 0.006). 

Local tumor progressions occurred in 17 patients despite grade A (absolutely curative) or B (relatively curative) judgments of their early responses. We reevaluated the ablative margins with a CT/MRI fusion imaging application, and then, 17 patients were finally categorized into grade C (margin-zero ablation, *n* = 8) and grade D (existence of residual HCC, *n* = 9). 

## 3. Discussion 

Our results show that some patients had a high risk of LTP, despite good judgements of their initial responses. In addition, we found that some residual lesions of HCC could be covered and obscured in the reactive hyperemic region. Thus, side-by-side comparisons and measurements of the ablative margin carry a risk of false and misleading diagnoses for LTP of HCC, even after multidisciplinary discussion. We considered the following three factors as risks in side-by-side comparisons: the image gap between pre- and post-ablation due to a distorted liver lobe by the ablation procedure; a misleading impression induced by a large non-enhancing area; and an unconscious desirability bias for positive outcomes. Overlaying pre- and post-RFA images can provide objective evidence to overcome such bias in subjective evaluations. 

Ablation therapy for HCC must include a sufficient margin of surrounding tissue to remove micrometastases and/or microvascular invasion. A previous study reported that micrometastases and microvascular invasion were not observed in well-differentiated HCC, while most moderate-differentiated HCCs less than 2.5 cm in diameter had them within 5 mm in the surrounding liver tissues [24,25,26]. A sufficient ablative margin also helps to overcome avoiding limitations due to partial volume effect on radiological assessment. Therefore, to be successful, RFA must not only eliminate arterial phase enhancement of HCC but also provide sufficient ablative margin. A safety margin of 5 mm or more is reportedly associated with a lower rate of LTP of HCC [27,28]. Meanwhile, remnant micrometastasis or microvascular invasion after ablation therapy may take months to grow into a detectable tumor. Several studies have reported the tumor-doubling time of HCC. Park et al. reported that the mean tumor volume and tumor diameter doubling times were 75 days (range, 21–209 days) and 219 days (range, 57–897 days), respectively [29]. According to another study, the median tumor volume doubling time of locally recurrent HCC after TACE was 69.7 days (range, 18–412 days) [30]. In the present study, some LTPs grew to approximately 1 cm in diameter at relatively short follow-up. Our data might support one main cause of early recurrent HCC being a residual tumor masked by ablative hyperemia, rather than growth from micrometastasis or microvascular invasion. 

Transient hyperemia surrounding the ablation zone appears immediately after RFA in most cases. Periablational hyperemia occurs with the inflammatory reaction to thermal injury, and pathology studies have revealed vascular dilation and granulation tissue in the inflamed area. Ablative hyperemia usually shows a poorly defined and thin rim of arterial phase enhancement on CECT or MRI, and sometimes partial thickness enhancement [31,32]. In fact, all nine HCCs regraded as D in this study were obscured by periablational hyperemia enhancements. Reactive hyperemia enhancement usually disappears by one month post-ablation, but can persist for several months [33,34]. 

Conventional CT/MRI fusion imaging applications were manually aligned by a complicated rigid-registration method with reference to intrahepatic landmarks [35,36,37,38]. The automatic tumor tracing and image registration were feasible and effective. These functions simplified the procedure of fusion registration and could enhance confidence in the treatment response assessment. Some mistracing and image gap could be encountered in Hepatic Guide; however, we consider that this CT/MRI overlay fusion imaging application can be used clinically because of the relatively low incidence. 

Our study had some limitations. First, this study was retrospective in design and had a small sample size; it is clinically challenging to investigate the unrecognized factors of locally recurrent HCC after RFA. Second, we used 5-mm image slices for reevaluation. Image slices can potentially be reconstructed at a thickness of 1.25 mm with our multi-detector CT, which would overcome the issue of partial volume effects. Further prospective comparative studies of this imaging technology with a larger number of patients are warranted. 

## 4. Materials and Methods

This study was approved by our institutional review board, and informed consent for percutaneous RFA was obtained from all patients. Diagnosis of HCC was based on the clinical guidelines of the American Association for the Study of Liver Disease following the observation of arterial hyperenhancement with washout on delayed-phase images [39,40]. 

### 4.1. Patients

This cohort study was conducted as a retrospective analysis of a prospective database in a single institution in which RFAs are routinely performed. The day after RFA therapy, dynamic contrast-enhanced CT or gadoxetic acid-enhanced MRI was performed, and ablative margin assessment of RFA was made with conventional side-by-side visual comparison of pre- and post-RFA images by consensus of the weekly multidisciplinary liver tumor board. The patients were classified into four groups as follows: grade A (absolutely curative), a 5-mm or larger ablative margin around the entire tumor; grade B (relatively curative), an ablative margin around the tumor but less than 5 mm in diameter in some places; grade C (margin-zero ablation), only an incomplete ablative margin around the tumor although no residual tumor was apparent; grade D (existence of residual HCC), the tumor was not completely ablated [14]. 

### 4.2. Equipment

Dynamic CT was performed with a 64-channel multi-detector CT (Discovery CT 750HD, GE Healthcare, Waukesha, WI, USA) at a slice thicknesses of 1.25 mm. All patients received 2.0 mL/kg of 300 mg I/mL nonionic contrast material (Iopamiron 300, Bayer-Schering Pharma AG, Berlin, Germany), which was intravenously injected over a fixed duration of 30 s using an automatic injector. Multiphasic CT was performed immediately before the contrast material injection, and at approximately 35–45, 65–80 and 190–205 s after an initiation of the injection for the arterial, portal and equilibrium phases, respectively. Gadoxetic acid-enhanced MR images were obtained on a 1.5-T MRI system (Signa HDx, GE Healthcare). For the enhancement study, a dose of 0.1 mL/kg of gadoxetic acid (Primovist 0.25 mmol/mL, Bayer-Schering Pharma AG, Berlin, Germany) was intravenously injected at a rate of 1.0 mL/s. Arterial phase timing was determined by the bolus-tracking method as the time at which the abdominal aorta reached peak time plus 13 s. The portal venous phase was scanned 20 s after the end of the arterial phase, and the transitional phase was scanned 100 s after the end of portal venous phase. Hepatobiliary phase imaging was performed 20 m after the injection. Our hospital adopts expedients such as CT/MRI with thin collimation and then storing CT images with 5-mm thickness at 5-mm intervals or MRI images with 6-mm thickness at 3-mm intervals to avoid excessive volume of data. 

Twenty cm-long, 17-gauge, monopolar internally cooled electrodes (VIVA RF ablation system; STARMed Co., Goyang, Gyeonggi, Korea) were used to deliver the radiofrequency energy. The active metallic tip can be adjusted in 5 mm intervals up to 3 cm, and the 200-W, 480-kHz monopolar radiofrequency generator regulates by impedance (VIVA RF generator, STARMed). Under Auto mode, power was initiated at 40 W with a 2-cm exposed-tip RF electrode or at 50 W with a 3-cm exposed-RF tip. All RFA procedures were performed by six experienced Hepatologists (M.T., H.C., M.T., S.H., Y.M. and H.I., with 7, 7, 13, 20, 21 and 22 years of experience, respectively). 

### 4.3. Assessment of Initial Technical Effectiveness 

The initial technical success of ablation was assessed based on dynamic contrast-enhanced CT or MRI findings the 24 h after RFA [41]. MRI was an option for patients with renal dysfunction or allergies to CT contrast material. The therapeutic response was evaluated by comparing axial images alone of the same modality obtained before and after RFA. A tumor was considered to have been successfully ablated when there were no enhanced regions within the entire tumor during the arterial phase and an ablative margin of normal liver tissue surrounding the tumor during the late arterial phase. Ablative margins were measured in a side-by-side manner. Part of the tumor was diagnosed as viable when images of the ablated area showed nodular peripheral enhancement [42,43]. 

The patients allocated grade A or B were discharged to outpatient follow-up, whereas the patients allocated grade C or D were performed additional RFA for obtaining sufficient ablative margin or destroying the residual portion the following week. 

### 4.4. Diagnosis of LTP 

LTP of HCC was diagnosed on the basis of the imaging findings in combination with the clinical findings including the tumor marker study. The term “local recurrence” includes both incompletely treated viable tumors and new tumors touching post-ablation necrosis [44]. Because recurrent HCCs after ablation tend to appear as hypervascular nodular lesions on the arterial phase [33], a crescent-like enhancing lesion touching the ablation zone was considered to be suspicious for LTP. In cases of indeterminate CT/MRI findings due to the small size of a lesion or an atypical enhancement pattern, short-term follow-up was performed for further evaluation or to determine possible interval growth of the lesion. If there were no nodular enhancement correlated with serum α-fetoprotein elevation, this was considered to be negative for recurrence. 

### 4.5. Image Fusion and Visualization of the Ablative Margin

Source data were archived on the hospital picture archive and communication system (PACS, Kodak, Rochester, NY, USA) in the standard DICOM (Digital Imaging and Communications in Medicine) format. Relevant datasets were retrieved from the PACS onto an image processing workstation (AW VolumeShare 7: 2.5 GHz processor, 32 GM RAM, GE Healthcare), equipped with the necessary software (Hepatic Guide, GE Healthcare). All fused images were generated by a radiologist (M.T.) and a hepatologist (T.M.). For image fusion, arterial phase images were used for the pre-RFA images, and late arterial phase for post-RFA images. 

Pre- and post-treatment images at 5-mm-slice thickness were coregistered on a workstation with Hepatic Guide installed. The first step is to establish a reference standard for pre-treatment HCC volumes. The HCC tumor contour was automatically traced in three dimensions after we drew a line along the axis of the tumor. When this tracing did not complete correctly, we could modify it manually in each image slice. After successful tumor tracing was confirmed, image registration could be carried out using both automated landmark-based least squares methods and automatic voxel-similarity method. The automatic registration adjustment function could be assigned to prioritize the tumor surroundings. When the gap of images was shown, we could modify it manually by referring to intrahepatic landmarks. After successful image registration was confirmed, the tumor image could be projected onto the ablation zone by an overlay of pre- and post-RFA images. The view could easily be switched to the axial, sagittal, or coronal plane to assess the three-dimensional ablative margin. The smallest distance between tumor border and margin of the ablation zone was defined as the minimal ablative margin. Image registration and fusion was processed on CT (*n* = 15) or MRI (*n* = 2) and interpreted by two hepatologists (T.M., Y.M.) with 7 and 21 years of experience in abdominal imaging. In cases of discrepancies between the two readers, a final decision was reached in a consensus session. The software allowed an image fusion on different modalities using CT/MRI; however, no patient was done comparing different imaging modalities in this study. 

### 4.6. Follow-Up and Data Analysis

If the follow-up images showed successful RFA and the absence of new tumors, dynamic contrast-enhanced CT or MRI were repeated at 2–4-month intervals. 

Data are expressed as the mean ± standard deviation (SD). The early response assessments 24 h after RFA were retrospectively rejudged by the use of fusion imaging application with automatic registration adjustment. Thereafter, we used the Mann–Whitney U test to test for differences in time to recurrence between the groups (re-grade C vs. re-grade D) using SPSS 22 (SPSS, Chicago, IL, USA). 

## 5. Conclusions 

The CT/MRI image fusion application could radiologically visualize the ablative margin by overlaying pre- and post-RFA images. The automatic image registration worked effectively on the workstation. Conventional side-by-side comparisons and measurements of the ablative margin can carry a risk of false and misleading diagnoses for local tumor progression of HCC. Overlay fused CT/MRI imaging with automatic registration can allow us to evaluate HCC ablative margin three-dimensionally with high accuracy. 

## Figures and Tables

**Figure 1 cancers-13-01460-f001:**
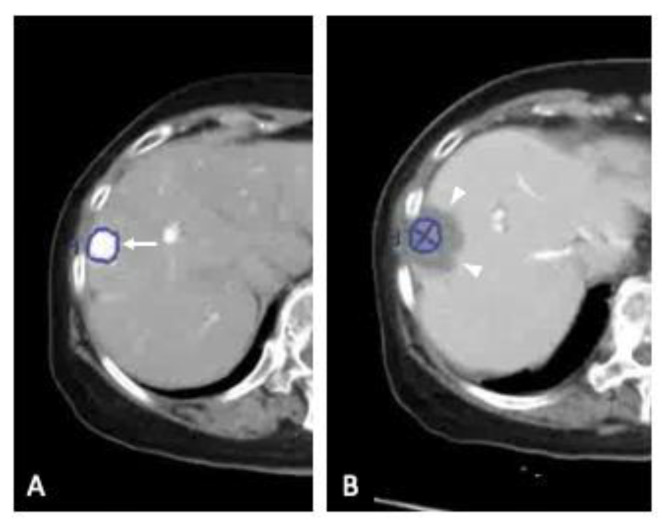
Good response after radiofrequency ablation (RFA). (**A**). Early-phase dynamic CT scan shows hepatocellular carcinoma (HCC) as an enhanced lesion (arrow) in segment 8 of the liver. The tumor border is marked with a thick blue line. (**B**). The traced tumor border is projected on to the low-density ablation zone (arrow heads) by the use of an overlay fused fusion application.

**Figure 2 cancers-13-01460-f002:**
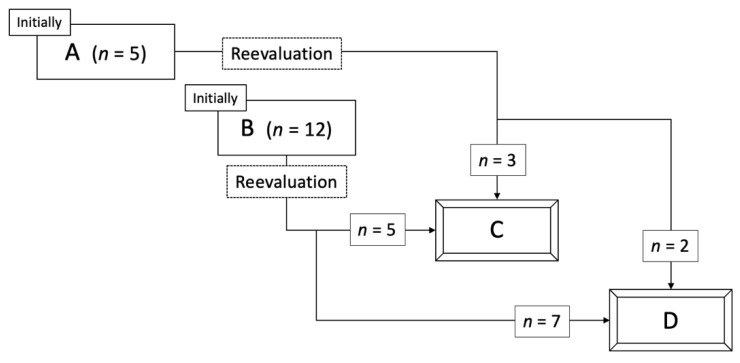
Changes in the classification of early treatment response reevaluation of RFA using overlay fused CT/MRI imaging.

**Figure 3 cancers-13-01460-f003:**
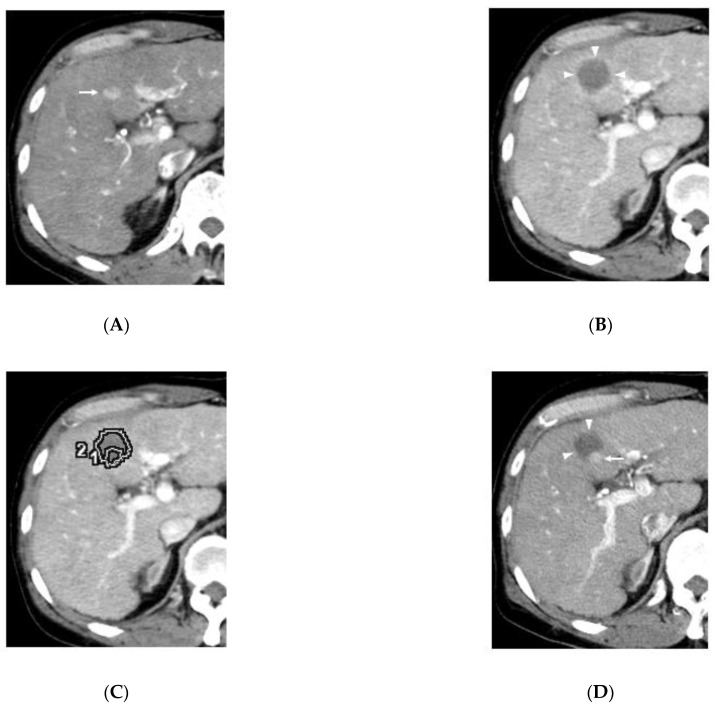
Residual hepatocellular carcinoma after RFA in a 65-year-old man in segment IV of liver. (**A**). Arterial phase CT shows HCC measuring 1.5 cm (arrow) before RFA. (**B**). Low density area due to RFA is extended (arrow heads) on late arterial phase CT, and then, the initial categorization was grade B (relatively curative). (**C**). The tumor border (circle 1) is fused into the image in Figure 2B (post RFA). A part of the tumor is located outside the low density area (circle 2) due to the RFA, meaning incomplete ablation. (**D**). Arterial phase CT obtained six months after the RFA shows an enhancing nodule measuring 1.0 cm (arrow) touching low density areas (arrowheads).

**Table 1 cancers-13-01460-t001:** Baseline Clinical Characteristics of Patients.

Characteristics	Values
Sex	
Male/Female	13/4
Age (year)	
Mean ± SD	68.3 ± 5.7
Range	58–79
Etiologic cause of HCC	
Hepatitis B/Hepatitis C/nonBnonC	2/12/3
Mean serum albumin level (g/dL) *	3.6 ± 0.4
Mean serum total bilirubin level (g/dL) *	0.9 ± 0.5
Child-Pugh class	
A/B/C	15/2/0
Serum AFP level (ng/mL)	
<20/20-200/>200	9/6/2
Number of HCCs	17
Tumor location	
Left lateral/Left medial/4/3/Right medial/Right lateral/Segment 1	4/3/5/5/0
Tumor size before ablation (cm)	
Mean ± SD	1.4 ± 0.5
Range	0.7–2.7
Coagulation size after ablation (cm)	
Mean LAD ± SD	3.2 ± 0.9
Range	1.6–5.4
Mean SAD ± SD	2.5 ± 0.5
Range	1.4–3.3

Unless otherwise stated, data are the number of patients or HCCs, HCC hepatocellular carcinoma, AFP alpha-fetoprotein, LAD long axis diameter, SAD short axis diameter. * Data are means ± standard deviation.

## Data Availability

The data presented in this study are available on request from the corresponding author.

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
