# Peer review of "Three-Dimensional Radiological Assessment of Ablative Margins in Hepatocellular Carcinoma: Pilot Study of Overlay Fused CT/MRI Imaging with Automatic Registration"

_cancers, 2021, doi:10.3390/cancers13061460_

Round 1
Reviewer 1 Report
- Figures 2a-2d need 3X zoom for residual hepatocellular carcinoma of single patient is note clear from current images particularly low density areas of 2b. Suggested to revise these images.
- What is the observed minimal ablative margin for RFA of HCC was not presented. Does the this > 2 and or < 5 cm needed to prevent local tumor progression. Provide the actual/observed ablative margin.
Author Response
We wish to express our appreciation to the reviewer for providing insightful comments on our paper. The comments have helped us significantly improve the paper.
- Figures 2a-2d need 3X zoom for residual hepatocellular carcinoma of single patient is note clear from current images particularly low density areas of 2b. Suggested to revise these images.
We replaced the case for Figure 2.
- What is the observed minimal ablative margin for RFA of HCC was not presented. Does the this > 2 and or < 5 cm needed to prevent local tumor progression. Provide the actual/observed ablative margin.
We are sorry that we missed to provide for an important information. We add an explanation of the observed minimal ablative margin. [Page 16, Line 10-12]
Thank you again for your careful review of our manuscript. We look forward to receiving your further response.

Reviewer 2 Report
Abstract:
- Suggest adding tumor characteristics to the introduction such as size.
- Did the 17 patient were the only one's who had LTP out of 467 patients? if the case, it is an extremely high success rate.
Introduction:
3. I would suggest removing the "recently" from page 1, line 48
4. Page 1 line 54, suggest changing "higher trends" to "higher rate". Studies have shown higher local progression for ablation compared to resection
Results:
5. Again, it is unclear if this 17 patients were the only ones who had LTP or these were out of the patients who were judged to have complete ablation then progressed. The authors need to clarify this. Would help to have an algorithm for the study population.
Author Response
We wish to express our appreciation to the reviewer for providing insightful comments on our paper. The comments have helped us significantly improve the paper.
- Abstract:
Suggest adding tumor characteristics to the introduction such as size.
We added the information of tumor size in keeping the word-count limit.
- Did the 17 patients were the only one's who had LTP out of 467 patients? if the case, it is an extremely high success rate.
It is generally single-digit percentage of local tumor progression for 2-3 years after RFA for HCC at Japanese high-volume center. Good results in Japan can be caused by our early detection and treatment according to the surveillance/follow-up program.
See the articles,
Shiina S, et al. Radiofrequency ablation for hepatocellular carcinoma: 10-year outcome and prognostic factors Am J Gastroenterol. 2012;107(4):569-577.
Waki K, et al. Percutaneous radiofrequency ablation as first-line treatment for small hepatocellular carcinoma: results and prognostic factors on long-term follow up. J Gastroenterol Hepatol 2010; 25: 597-604
- Introduction:
I would suggest removing the "recently" from page 1, line 48
We removed the word, “recently”.
- Page 1 line 54, suggest changing "higher trends" to "higher rate". Studies have shown higher local progression for ablation compared to resection
We revised this sentence.
- Results:
Again, it is unclear if this 17 patients were the only ones who had LTP or these were out of the patients who were judged to have complete ablation then progressed. The authors need to clarify this. Would help to have an algorithm for the study population.
Generally, initial judgments of successful ablation are done in Japan referring to contrast-enhanced images obtained in the 24 hoursafter ablation. Then, we had some cases with residual HCC that was covered in the reactive hyperemic region in this study.
We emphasized in Discussion that side-by-side comparisons and measurements of the ablative margin can carry a risk of false and misleading diagnoses for LTP of HCC even after multidisciplinary discussion, pointing the three factors.
We modified the section of “Assessment of initial technical effectiveness”.
Thank you again for your careful review of our manuscript. We look forward to receiving your further response.

Reviewer 3 Report
The work reports a retrospective study to evaluate the margins of ablation by RFA for HCC. The authors investigate the clinical feasibility of employing fusion computed tomography/magnetic resonance imaging (CT/MRI) to assess accurately the ablation margins in RFA for HCC.
Though the study was performed with a minimal number of patients (17), it is consistently reported as a pilot study.
The subject of the manuscript is interesting; however, as CT/MRI fusion imaging approach is routinely used for assessing hepatic lesions, the novelty detected is restricted to the application of CTI/MRI methods for determining the ablation margins of HCC in RFA, which should increase the accuracy of the ablation.
Major points:
To make the manuscript more impactful and convince the scientific community to adopt CT/MRI fusion approach for evaluating the margins of RFA, the authors should analyze the accuracy of ablation margins of HCC of their cohort of patients more quantitatively. To this aim, the author should support and implement their results by reporting more axial scan images pre- and post-ablation from several patients (and possibly inserting video in the supplementary materials section) of side-by-side evaluations and the proposed fused approach. Also, the authors could describe more in details, how a novel system of coordinate will define more accurately the margins of ablation through CT/MRI fusion approach.
In the methods, details on the statistical analysis used are not provided.
Minor points:
-The legends of figures should be more explicative.
-The quality of the images should be improved.
Author Response
We wish to express our appreciation to the reviewer for providing insightful comments on our paper. The comments have helped us significantly improve the paper.
- To make the manuscript more impactful and convince the scientific community to adopt CT/MRI fusion approach for evaluating the margins of RFA, the authors should analyze the accuracy of ablation margins of HCC of their cohort of patients more quantitatively. To this aim, the author should support and implement their results by reporting more axial scan images pre- and post-ablation from several patients (and possibly inserting video in the supplementary materials section) of side-by-side evaluations and the proposed fused approach. Also, the authors could describe more in details, how a novel system of coordinate will define more accurately the margins of ablation through CT/MRI fusion approach.
We modified the section of Materials and Methods.
If we could not satisfy you, we are ready to receive your further response.
- In the methods, details on the statistical analysis used are not provided.
We modified the section of “4.6. Follow-up and data analysis”.
- Minor points:
The legends of figures should be more explicative.
The quality of the images should be improved.
We replaced the case for Figure 2.
Thank you again for your careful review of our manuscript. We look forward to receiving your further response.
